# The Challenges of Children with Bipolar Disorder

**DOI:** 10.3390/medicina57060601

**Published:** 2021-06-11

**Authors:** Robert M. Post, Heinz Grunze

**Affiliations:** 1School of Medicine, George Washington University, Washington, DC 20052, USA; 2Bipolar Collaborative Network, 5415 W. Cedar Lane, Suite 201-B, Bethesda, MD 20814, USA; 3Psychiatrie Schwäbisch Hall, Campus ZfPG, 74523 Schwäbisch Hall, Germany; heinz.grunze@icloud.com; 4Campus Nuremberg-Nord, Paracelsus Medical University, 90419 Nuremberg, Germany

**Keywords:** anticonvulsants, antipsychotics, bipolar disorder, children, early recognition, family focused therapy, lithium

## Abstract

Childhood onset bipolar disorder (CO-BD) presents a panoply of difficulties associated with early recognition and treatment. CO-BD is associated with a variety of precursors and comorbidities that have been inadequately studied, so treatment remains obscure. The earlier the onset, the longer is the delay to first treatment, and both early onset and treatment delay are associated with more depressive episodes and a poor prognosis in adulthood. Ultra-rapid and ultradian cycling, consistent with a diagnosis of BP-NOS, are highly prevalent in the youngest children and take long periods of time and complex treatment regimens to achieve euthymia. Lithium and atypical antipsychotics are effective in mania, but treatment of depression remains obscure, with the exception of lurasidone, for children ages 10-17. Treatment of the common comorbid anxiety disorders, oppositional defiant disorders, pathological habits, and substance abuse are all poorly studied and are off-label. Cognitive dysfunction after a first manic hospitalization improves over the next year only on the condition that no further episodes occur. Yet comprehensive expert treatment after an initial manic hospitalization results in many fewer relapses than traditional treatment as usual, emphasizing the need for combined pharmacological, psychosocial, and psycho-educational approaches to this difficult and highly recurrent illness.

## 1. Introduction

Epidemiological data by Merikangas et al., (2010) indicated that 2.2% of adolescents in the US have a bipolar spectrum disorder, including BP-NOS, but disappointedly only 20% are in any kind of treatment [1]. Other data indicate a similar percentage of bipolar I children in other countries, but if those with BP-NOS are considered, the incidence may be considerably higher, perhaps around 5%. Childhood onsets of bipolar disorders are more common in the US than in many European countries with one quarter of onsets in adults with bipolar disorder occurring before age 13 and two thirds before age 19 [2,3,4,5] Multiple factors account for this; among the most prominent are an increased incidence of a positive family history of mood disorders and substance abuse disorders in patients’ parents and grandparents [6,7,8] and an increased incidence of multiple psychosocial adversities (different types of abuse) in childhood [9]. There are additive effects on earlier age on onset when there is the combined presence of both a high loading of family history and incidence of childhood adversity [7]. Other risk factors in the US include obesity, an inflammatory diet, and poor access to health care yielding longer delays to diagnosis and first treatment.

## 2. Transgenerational Transmission and Illness Evolution

Not only does bipolar disorder in adults in the US compared to Europeans appear to be a more pernicious illness [2], but compared to the Europeans, the children of US adults with bipolar disorder likely have an increased incidence of bipolar disorder, depression, substance abuse, suicidality, and other psychiatric illnesses [3]. The data are convergent with those of Axelson et al. [4], indicating that upon a systematic 8 years of follow up, children of a bipolar parent compared to a control parent in the community are at increased risk (in descending order) for an anxiety disorder, depression, disruptive behavioral disorder, ADHD, and substance abuse, each more than the approximately 20% who will develop a bipolar disorder diagnosis.

Considerable data also suggest that there may be an evolution in illness manifestations in those at high risk because of a parent with bipolar disorder from the initial appearance of an anxiety disorder to a depression and then to bipolar disorder [5]. In US cohorts this may be associated with a high risk of multiple early prodromes, including anxiety, ADHD, ODD, as well as depression and then bipolar disorder [10], and each of these may show an increasing incidence from a cohort effect [6].

## 3. Early Recognition and Treatment

Early recognition and treatment are critical as childhood onset illness as compared to adult onset is associated with a problematic course of increased substance abuse, suicide, and episodes of mood disorder in adulthood [7,8]. Lack of aggressive comprehensive treatment is associated with increase in risk not only of relapses [9] and social and educational dysfunction, but also of cognitive dysfunction if further episodes occur in the year after a first mania [11,12]. Attempts to enhance cognitive reserve deserve further study and consideration.

However, the ideal treatment of childhood onset bipolar disorder is compromised by a lack of systematic clinical trials, and current guidelines that are largely based on expert opinion. Children are being treated with a high incidence of atypical antipsychotics, but lesser degrees of lithium and mood stabilizing anticonvulsants, such as valproate, lamotrigine, or carbamazepine. Yet naturalistic follow up studies suggest that those treated with lithium have better long-term outcomes and more euthymic days [13] or fewer days of feeling depressed and less suicidality [14] than other treatment and mood stabilizer approaches.

### 3.1. The Case for Lithium Treatment of Children

Lithium has positive controlled data for treatment of acute mania [15] and prevention [16]. After a first manic hospitalization, Berk et al. [17] reported that one year of randomized treatment with lithium was superior to that of quetiapine on measures of mania, depression, functioning, cognition, and brain imaging. In addition, lithium has multiple assets that have not yet been systematically addressed in children [18]. Lithium increases hippocampal volume in adults, likely based on its ability to increase BDNF and neurogenesis. It also increases white matter integrity in children who have been shown to have these deficits. In adults, lithium helps prevent cognitive deterioration in patients with mild cognitive impairment and it reduces the incidence of a diagnosis of dementia in old age [19,20]. Based on its ability to stimulate the enzyme telomerase, lithium protects the length of telomeres, which are reduced in length by stress and depression [21]. This and other diverse mechanisms could account for the data suggesting that lithium could help prevent a variety of medical illnesses including cancer [22].

Lithium appears most effective in those with a positive family history of mood disorders and especially if there is a positive family history of response to lithium. In addition, those with classic presentations of euphoric mania, clear well-intervals between episodes, a lack of anxiety or substance abuse comorbidity, and non-rapid cycling are more likely to respond [23]. Initiating lithium treatment early in the course of illness is also more productive than latter treatment after multiple episodes or rapid cycling has occurred [24,25,26]. Lithium is FDA- and EMA-approved for children over 12 years of age.

### 3.2. Atypical Antipsychotics

While some atypical antipsychotics are FDA-approved for preventing mania and total episodes, some, such as aripiprazole, have not shown significant efficacy in preventing depressions. Ziprasidone is effective in mania, but it failed in depression in adults. Olanzapine has efficacy in mania and depression, but issues of weight gain and metabolic tolerability make it a third line option [27]. Those with a parental history of non-response to lithium maybe more likely to respond to atypicals [28].

Lurasidone has efficacy in bipolar depression in children aged 10–17 years as well as in adults, but long-term preventive data are mixed [29]. It is of interest that in studies of bipolar depression in both children and adults, lurasidone was more effective and had a bigger effect size in those with baseline elevations in the inflammatory marker C-reactive protein (CRP) [30,31].

### 3.3. Mood Stabilizing Anticonvulsants

As opposed to robust data in adults with mania, data on valproate and carbamazepine in mania in children are more mixed and equivocal. Valproate is widely used in mania in children, although several placebo-controlled trials failed to show efficacy in acute mania. It should not be used in women of childbearing age [32]. Oxcarbazepine (OXC) was not effective in the whole group of children with bipolar disorder, while the youngest compared to the oldest did significantly better on OXC than placebo [33]. Data showed the opposite pattern on lamotrigine where older patients fared better than younger children [34]. As noted above, in the long-term naturalistic studies those treated with lithium performed better than those treated with atypicals and anticonvulsants [14].

### 3.4. Family History Is Useful in the Choice of Mood Stabilizing Treatment

Considerable evidence indicates that a positive family history of mood disorders is a predictor of response to lithium [23]. Conversely, those without a family history of bipolar disorder are more likely to respond to carbamazepine (along with characteristics that are the mirror image of lithium non-response, such as more continuous cycling and the presence of anxiety and substance abuse comorbidity, and the presence of mood incongruent delusions). Interestingly, those with a personal or family history of anxiety disorders are more likely to respond to lamotrigine [35].

### 3.5. Combinations Are More Effective than Monotherapy

Findling et al. [36] reported that children stabilized on the combination of lithium and valproate relapsed at a very high rate when they were randomized to either monotherapy. They then rapidly re-responded when combination therapy was re-instituted indicating the need for combination therapy in the majority of children with bipolar disorder. Geller et al. [37] came to the same conclusion in her randomized study of lithium, valproate, and risperidone, in that most patients required a combination of the atypical with lithium or valproate (and often with adjunctive stimulants) to achieve adequate mood stabilization. Kowatch et al. [38] also concluded that combinations were ultimately needed in his randomized study of lithium, valproate, and carbamazepine in acute mania.

Given the high incidence of comorbid ADHD and bipolar disorder in children, most investigators recommend first achieving mood stabilization prior to the use of stimulants for residual symptoms of ADHD. Adding a stimulant in this fashion is not associated with exacerbation of the bipolar disorder [39]. First treating these comorbid patients with high dose stimulants or antidepressants prior to mood stabilization risks a high incidence of non-response or a worsening of symptoms [40].

### 3.6. Other Agents and Supplements for Comorbidities

Some drugs that are not effective antimanic agents, such as gabapentin, still have evidence of effectiveness in other syndromes and comorbidities, including alcohol use disorder (AUD), avoidance and anxiety disorders, and anxiety attacks, such as topiramate and zonisamide in AUD and bulimia [41]. Two supplements available without prescription have a wide range of potential benefits in bipolar disorder as listed in Table 1. For example, N-acetylcysteine (NAC) has placebo-controlled data for effectiveness in depression and anxiety in adult bipolar patients, as well as the ability to limit multiple drugs of abuse and habits such as gambling, OCD, and trichotillomania [41,42,43]. There are three positive placebo-controlled trials of NAC for irritability in autism in children 4–17 years of age so that safety in children has been demonstrated.

Acetyl-L-carnitine (LAC) has been reported low in the blood of adult depressed patients, especially those with early onset, more severe and treatment refractory depression, and in those with histories of abuse in childhood [44]. It has antidepressant effects in patients and in animal models of depression, but effectiveness has not yet been demonstrated in children. However, the possibility of normalizing LAC blood levels in depressed patients with a history of adversity in childhood suggests the importance of studying this subgroup as patients with bipolar disorder have such a high incidence of histories of abuse in childhood which in turn is associated with relative treatment refractoriness. LAC also has anti-pain effects, sensitizes insulin receptors, and improves peripheral neuropathy [44,45].

Other agents not listed in Table 1 also deserve consideration even though the data on effectiveness in children is not solid [41,46]. For example, there is a very high incidence of borderline or low levels of vitamin D3 in children with major psychiatric illness, so supplementation with this safe vitamin would appear worthy of consideration. Similarly folic acid has been reported to enhance the effectiveness of lithium and many antidepressants, and for those with a methylenetetrahydrofolate reductase (MTHFR) deficiency, supplementation with L-methylfolate is a necessity.

This and other suggestions for addressing some of the complexities of bipolar disorder are acknowledged to be inadequately supported in the literature sufficiently to merit routines use [46]. However, given the paucity of systematic studies in children and the wide range of impairing symptoms and syndromes seen in the illness [18,47], some consideration of using these agents on the basis of careful evaluation of their potential risk: benefit ratio may be a rational approach. The potential role of the calcium channel blocker nimodipine also deserves further evaluation and consideration [23,48,49].

## 4. Psychotherapy: The Case for Systematic Employment of Family Focused Therapy and Similar Therapies

While optimal pharmacological approaches to childhood onset bipolar disorder remain n controversial, there is much consensus about the importance of psycho-social therapies.

Recent meta-analyses document the benefit of family focused therapy (FFT) and related group therapies [50]. Miklowitz et al. [51] also found that FFT was effective for those with prodromal symptoms including bipolar disorder not otherwise specified (BP-NOS) for producing stabilization in depression, reducing recurrence risk, and increasing psychosocial functioning, each with moderate to large effect sizes. Pavuluri et al. [52] endorsed family focus cognitive behavioral therapy, and Fristad et al. [53] also found multifamily psychoeducational therapy highly effective.

In addition, a recent report indicates that cognitive behavioral therapy (CBT) can be modified for transdiagnostic use in children and is more effective that treatment as usual [54]. Together these data provide strong evidence for the utility of FFT and other family interventions, and CBT for those with early symptoms, prodromes, and more full-blown bipolar disorder. Given the efficacy and safety of these psychosocial approaches and the utility of psycho-educational efforts, it would appear imperative that these types of therapies be utilized adjunctively in the routine treatment of early affective disorders in children.

## 5. The Need of Psychological and Pharmacological Attempts to Head of Early Illness in Those at Highest Risk

Given the high risks for the onset of diverse childhood psychiatric diagnoses in children of a parent with bipolar disorder [4], attempting a primary prevention would appear indicated. Endorsing attempts to achieve a good diet, regular exercise, sports participation and sleep habits, and mindfulness/meditation have not only been proven to be effective, but their safety, practicality, and theoretical rationale make them highly recommended [43]. As noted above, family therapy and CBT have considerable support in systematic studies.

A variety of safe pharmacological interventions may have merit in primary and/or secondary prevention even if not fully validated by systematic studies [46]. Along with safety, the non-specificity of potential effectiveness across multiple symptoms and syndromes makes their early utilization in children worthy of consideration. These could include omega-3-fatty acids, N-acetylcysteine, minocycline, and other anti-inflammatory drugs (especially in the presence of positive markers of inflammation), deserving further study and perhaps individual clinical trials. In those with a history of adversity in childhood, one might even consider a clinical trial with acetyl-L-carnitine (LAC) since levels of LAC in the blood of adult depressed patients are especially low in those with a history of childhood adversity [44,45].

## 6. Conclusions

The majority of follow up studies indicate that childhood onset bipolar disorder is not a benign illness and patients remain ill some two thirds of the time and experience considerable dysfunction [13,55,56]. Suicide and substance abuse remain a too common liability. Given the fact that the number of prior episodes is associated with the degree of cognitive dysfunction and the development of treatment resistance, preventing episodes from the outset become a major goal of treatment [46,57,58]. However, given the paucity of systematic research to guide optimal therapeutics, the complexity of presentations, and the need for comprehensive pharmacological and psychosocial intervention, achieving a good long-term outcome is not readily accomplished. This is all-the-more an imposing task as the majority children treated for psychiatric illness in the US are primary care physicians who are typically not well versed in the nuances of diagnosis and treatment [59].

Thus, there is a major need for more treatment research to better guide both psychiatrists and primary care physicians. Psychotherapeutic and psycho-educational are a necessary part of treatment and the limited access to expert referral is an additional problem that must be overcome. Since there are so many obstacles to achieving excellent treatment outcomes, it behooves the clinician to request help from parents in facilitating the diagnosis and the assessment of response to treatment by getting weekly rating of their child’s symptoms [47,60,61]. Parents can receive confidential IRB-approved emails every Sunday for rating the severity of anxiety, depression, attention deficit hyperactivity disorder (ADHD), oppositional behavior, and mania which can then be printed out for ease of longitudinal assess by the physician. Informed consent, demographic assessment, and receiving the ratings can be acquired at http://bipolarnews.org/?page_id=2630 (last accessed on 10 June 2021). With creative comprehensive treatment, often involving lithium, multiple other agents, and adjunctive psychotherapy, life-saving outcomes can be achieved in even the very difficult occurrences of childhood onset bipolar disorder and its many comorbidities.

## Figures and Tables

**Table 1 medicina-57-00601-t001:** Potential utility of supplements and drugs in the treatment of comorbidities of bipolar disorder. Note that all are off label in bipolar disorder except valproate *.

N-Acetylcysteine (NAC)	Acetyl-L-Carnitine (ACL)	Valproate (VPA)	Topiramate	Zonisamide	Gabapentin	Modafinil
Cocaine	Pain	Migraine	Alcohol	Alcohol	Anxiety	Cocaine
Alcohol	Insulin-receptor sub-sensitivity	Anxiety	Cocaine	Bulimia	Social phobia	Narcolepsy
Nicotine	Peripheral neuropathy	Alcohol	Bulimia		Alcohol	ADHD
Marijuana	Hypertension		Migraine		Pain	
Gambling			Anger attacks			
OCD						
PTSD						
Trichotillomania						
Depression						
Autism						

* References cited in [41].

## Data Availability

All data given in this review have been previously published and are in the public domain.

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
