# Peer review of "The Challenges of Children with Bipolar Disorder"

_medicina, 2021, doi:10.3390/medicina57060601_

Round 1

Reviewer 1 Report

The manuscript is a short but clear and complete update on the treatment of bipolar disorder in children. The Authors correctly stress that the guidelines are essentially based on experts’ opinions being the trials on this topic scant.

The paper is adherent to the journal standards and well-organized even if more as a chapter of a book than as a paper.

The content is of high quality and provides useful information for clinicians to face a challenging topic: children with bipolar disorder.

In my opinion the Authors should review the paper according to the guidelines of the Journal (Introduction, Materials and Methods, Results, Discussion and Conclusion).

Furthermore, Authors should report the current limitation on the use of the lithium in section 3.1 (children over 12 years old) and of valproate in section 3.3 (no prescription in girls).

Author Response

The way the paper is laid out would not be amenable to the formatting as requested as each section is a mixture of multiple of these headings.

The comments on lithium and valproate have been added

Reviewer 2 Report

This is a very useful, well-written and summarized review on the challenges of Childhood onset bipolar disorder. I’d like to congratulate the authors for their great work. Please find some minor comments:

Minor comments:

  1. “Not only is bipolar disorder in the US a more pernicious illness [1], but compared to the Europeans, the children of US adults with bipolar disorder have an increased incidence of not only bipolar disorder, but depression, substance abuse, suicidality, and other psychiatric illnesses [2].”

Maybe tone down this sentence based on a multicentric questionnaire.

  1. “Epidemiological data indicate that 2.2% of adolescents in the US have a bipolar spectrum disorder, but disappointedly only 20% are in any kind of treatment [1]. Other data indicate a similar percentage of bipolar I children in other countries, but if those with BP-NOS are considered, the incidence may be considerably higher, perhaps around 5%.”

Please elucidate whether BP-NOS has been considered within the bipolar spectrum in the prevalence study.

  1. In the section “Early Recognition” it may be interesting to point out the concept of “cognitve reserve” (https://pubmed.ncbi.nlm.nih.gov/31035381/), which has been recently incorporated in psychiatry and is of particular importance in childhood onset illness. Regarding early recognition, it is key to perform close monitoring the offspring of patients diagnosed with BD. The enhancement of the cognitive reserve in this collective at risk may prove to be of utmost importance in the near future (https://pubmed.ncbi.nlm.nih.gov/33631372/).

  1. Reference 18 could be updated with a recent meta-analysis: https://onlinelibrary.wiley.com/doi/abs/10.1111/acps.13153

  1. You could maybe add that the increased risk of cancer in BD may be reduced with lithium treatment, due to its multiple beneficial biochemical properties: https://pubmed.ncbi.nlm.nih.gov/33831461/

  1. Most references (16/57) included in the review are from the same author. This is a limitation for the generalization of the arguments exposed.

For example: “The potential role of the calcium channel blocker 196 nimodipine also deserves further evaluation and consideration [20,44,45].”

This categoric argument should be supported with more evidence from scientific studies (either clinical or animal studies).

Author Response

1.  We have toned down and softened this statement.  2.  BP-NOS was included in the spectrum. 3. A comment on cognitive reserve has been added.  4. Updated reference included; thank you.  5. Statement on cancer and other medical illness has been added.  6.  The citations on nimodipine cover much work in adults and a case study in an ultradian cycling adolescent.  However, the preliminary nature of these data are indicated by the statement that further study is required.